# Efficacy of Dexmedetomidine vs. Remifentanil for Postoperative Analgesia and Opioid-Related Side Effects after Gynecological Laparoscopy: A Prospective Randomized Controlled Trial

**DOI:** 10.3390/jcm12010350

**Published:** 2023-01-02

**Authors:** Jung Min Koo, Youn-Jee Chung, Mihyeon Lee, Young Eun Moon

**Affiliations:** 1Department of Anesthesiology and Pain Medicine, Seoul St. Mary’s Hospital, College of Medicine, The Catholic University of Korea, Seoul 06591, Republic of Korea; 2Department of Obstetrics and Gynecology, Seoul St. Mary’s Hospital, College of Medicine, The Catholic University of Korea, Seoul 06591, Republic of Korea

**Keywords:** intraoperative analgesia, dexmedetomidine, opioid-related side effects, remifentanil

## Abstract

Remifentanil is widely used for intraoperative analgesia, but often causes remifentanil-induced hyperalgesia (RIH) and related side effects. Dexmedetomidine, a non-opioid analgesic, has been used as an alternative to remifentanil to prevent RIH. We aimed to investigate the effect of dexmedetomidine on postoperative recovery after gynecological laparoscopy. Ninety-six adult patients undergoing elective gynecological laparoscopy were randomly assigned to the dexmedetomidine or remifentanil groups. The primary outcome was the pain score at 30 min after surgery. The secondary outcomes were intraoperative adverse events (hypotension and bradycardia) and postoperative opioid-related side effects (nausea, vomiting, requirement for rescue analgesics, and shivering). We also performed an ancillary cytokine study to evaluate oxidative stress, one of the causes of RIH. Compared with the remifentanil group, the dexmedetomidine group had lower pain scores at 30 min after surgery (4.0 ± 1.9 vs. 6.1 ± 2.0, mean ± SD, *p* < 0.001) and lower incidence of intraoperative hypotension and postoperative nausea, vomiting, and shivering. Furthermore, the proportion of patients requiring rescue analgesics was significantly lower in the dexmedetomidine than in the remifentanil group (25% vs. 66.7%, *p* < 0.001). Cytokine levels did not differ between the groups. Dexmedetomidine showed a better analgesic effect with minimal opioid-related side effects and is considered superior to remifentanil for intraoperative analgesia.

## 1. Introduction

Remifentanil has gained popularity as an adjunct during general anesthesia and is widely used for pain control during and after surgery. However, recent studies have revealed that remifentanil is associated with postoperative opioid-induced hyperalgesia, along with disturbances in the immune and inflammatory systems. Opioids are known to trigger opioid-induced hyperalgesia, particularly short-acting ones. Remifentanil induces rapid tolerance, even after only 90 min of infusion [1]. As a result, remifentanil-induced hyperalgesia (RIH) leads to considerable postoperative opioid consumption and is accompanied by opioid-related complications, such as nausea, vomiting, respiratory depression, and ileus [2,3]. The oxidative stress theory is one explanation of the mechanisms underlying RIH during the perioperative period. Several animal studies have demonstrated that remifentanil induces overproduction of reactive oxygen species, which directly activate N-methyl-D-aspartate (NMDA) receptors and matrix metalloproteinase (MMP)-9. In turn, upregulation of MMP-9 reactivates NMDA receptors, causing central hypersensitivity, which clinically manifests as hyperalgesia [4,5,6].

In the last decade, dexmedetomidine has been increasingly used in perioperative clinical settings owing to its sedative and analgesic effects. This centrally acting non-opioid analgesic is known to reduce surgical stress responses [7,8,9,10]. Recent animal studies showed that dexmedetomidine reduces NMDA receptor reactivity and MMP-9 production [11,12]. However, there have been few clinical studies of its effects, particularly in female patients undergoing gynecological laparoscopy.

Therefore, the aim of our study was to compare the ability of dexmedetomidine and remifentanil to reduce postoperative pain, opioid consumption, and opioid-related side effects after gynecologic laparoscopy. Furthermore, we measured MMP-9 and manganese superoxide dismutase (MnSOD) to determine whether dexmedetomidine can reduce postoperative oxidative stress responses.

## 2. Materials and Methods

### 2.1. Study Design & Participants

This prospective, parallel-group, single-blind, randomized controlled trial was conducted in a tertiary university hospital in South Korea. The study population included female patients aged 20–65 years who were scheduled for elective gynecological laparoscopy, including hysterectomy, oophorectomy, salpingectomy, cystectomy, cyst enucleation, and uterine myomectomy. Patients with American Society of Anesthesiology (ASA) classification III or more, those scheduled for cancer surgery, those undergoing emergency surgery, and those with chronic pain on medication, history of psychiatric diseases, hypotension, bradycardia, atrioventricular block, intraventricular or sinoatrial block, known allergies to any of the study drugs, pregnancy, or lactation were excluded from the study.

### 2.2. Randomization and Blinding

Patients enrolled in the study were randomly assigned to the remifentanil (group R) or dexmedetomidine group (group D). Random allocation was performed according to the numbers generated by the computer. Once a patient was enrolled, the medical staff opened the opaque, sequentially numbered envelope containing the group allocation.

Patients and surgeons were blinded to the group allocation throughout the study period. Healthcare providers providing postoperative care and those measuring postoperative outcomes were unaware of the group allocation. The anesthesiologist performing the general anesthesia could not be blinded because of the different drug infusion protocols between the groups. However, they did not participate in the postoperative care or the assessment of postoperative outcomes.

### 2.3. Intervention

On the day before surgery and upon entering the operating room, patients were educated on the concept of the numeric rating scale (NRS). The NRS is an 11-point rating scale for pain, with 0 indicating no pain and 10 indicating the worst possible pain. Patients were asked to verbally indicate their current pain status on a scale of 0 to 10. They were also instructed to request analgesic medication from the healthcare providers if the NRS pain score exceeded 4.

No premedication was administered before anesthesia induction. Monitoring was initiated on arrival to the operating room, including electrocardiography, blood pressure measurement, pulse oximetry, neuromuscular monitoring using train-of-four (TOF) stimulation, and bispectral index.

Anesthesia was induced with a bolus intravenous injection of 1.5–2 mg/kg propofol. In the absence of eyelash reflex, 0.8 mg/kg of rocuronium was injected. After TOF reached 0, orotracheal intubation was performed. Anesthetic maintenance with desflurane at 4 L/min flow to reach an expired concentration of 4–6% in a mixture of 40% air-to-oxygen ratio was continued until completion of surgery. Mechanical ventilation was maintained with a tidal volume of 4–6 mL/kg to reach an end-tidal CO_2_ of 25–40 mmHg. Additional rocuronium was administered if an additional neuromuscular block was required. During the surgery, ephedrine 4 mg was injected if the systolic blood pressure (SBP) was <80 mmHg or if the mean arterial pressure (MAP) was <60 mmHg. If the heart rate (HR) dropped below 45 bpm, atropine (0.25–0.5 mg) was administered.

For the patients in group D, a loading dose of dexmedetomidine was administered before anesthesia induction at 0.7 μg/kg for 10 min and followed by a continuous infusion of 0.5 μg/kg/h. Infusion rate was adjusted in steps of 0.1 μg/kg/h to maintain the SBP within ±20% of baseline. Dexmedetomidine was discontinued at completion of surgery.

For the patients in group R, remifentanil 1.5 μg/kg was administered before anesthesia induction and followed by a continuous infusion of 0.15 μg/kg/min. Infusion rate was adjusted in increments of 0.02 μg/kg/min to maintain the SBP within ±20% of the baseline. Remifentanil was discontinued at completion of surgery.

Laparoscopy was performed under video guidance, with three punctures in the abdomen. The intraperitoneal pressure was maintained at approximately 12 mmHg. To prevent postoperative nausea and vomiting (PONV), the attending anesthesiologist injected intravenous dexamethasone (5 mg) and palonosetron (75 µg) at the beginning and end of surgery, respectively. Intravenous acetaminophen (1 g mixed with 100 mL of 0.9% normal saline) was infused 30 min before the end of surgery. Endotracheal tube extubation was performed when the TOF was 4, and the patients were sent to the recovery room.

During the recovery period, intravenous fentanyl 0.5–1 μg/kg was immediately administered if the NRS scores were 4 or higher. The regimen of intravenous patient-controlled analgesia (AutoMed 3200; Ace Medical, Seoul, Republic of Korea) was fentanyl 15 μg/kg in normal saline 100 mL at a basal rate of 0 mL/h (no basal rate of infusion), bolus of 1 mL (0.15 μg/kg), and lock-out time of 10 min. When patients experienced PONV, alleviation was attempted with 10 mg of metoclopramide.

### 2.4. Outcome Measurement

To compare the hemodynamic effect of dexmedetomidine with that of remifentanil, we recorded the incidence of intraoperative bradycardia (HR < 60/min) and hypotension (SBP < 80 mmHg or MAP < 60 mmHg). Additionally, the time to eye opening and time to extubation from discontinuation of the volatile anesthetic were also measured.

The primary outcome was the NRS pain score at 30 min after surgery. Pain intensity was assessed using the NRS upon arrival at the recovery room and every 15 min thereafter. The level of sedation (awake/sedated and responsive to verbal stimuli/sedated and unresponsive to verbal stimuli) was assessed 30 min after the end of surgery, and the time spent in the recovery room was recorded. Moreover, any events of PONV and shivering, and additional analgesic or antiemetic requirements were recorded. These outcomes were evaluated again 24 h after surgery. Finally, the time to first defecation or flatus was recorded before discharge from the ward.

To investigate the effect of dexmedetomidine on oxidative stress, an ancillary cytokine study was performed on 40 patients (20 from each group). MnSOD and MMP-9 levels were measured at baseline, immediately after surgery, and 24 h after surgery. All blood samples were placed in pre-chilled tubes on ice and centrifuged within 60 min. Plasma from the samples was separated and stored at −70 °C until analysis. These measurements were made using enzyme-linked immunosorbent assay kits for MnSOD (Mybiosource, San Diego, CA, USA.; sensitivity: 0.055 ng/mL, detection range: 0.156–10 ng/mL) and MMP-9 (ThermoFisher, Waltham, MA, U.S.A.; sensitivity: 0.2 pg/mL, detection range: 0.99–4050 pg/mL).

### 2.5. Statistical Analysis

The sample size was calculated with reference to a pilot study of laparoscopic gynecological surgery using remifentanil as an intraoperative opioid. The mean ± standard deviation of NRS in the recovery room during this pilot study was 6.3 ± 3.2. Considering a 30% reduction in pain scores from 6.3 to 4.4, the appropriate sample size was 45 for each group, with power (1-β) = 0.8, and α (2 sided) as 0.05. Considering a dropout rate of 10%, a total of 100 patients were planned to be enrolled, with 50 patients in each group.

Data distribution was examined using the Kolmogorov–Smirnov test. The independent sample *t*-test or Mann-Whitney’s *U* test was used for comparison of quantitative variables, and Pearson’s χ^2^ test or Fisher’s exact test was used to analyze categorical data. Kaplan–Meier analysis was performed to compare the time of fentanyl administration and the proportion of patients not receiving fentanyl in the recovery room. Repeated-measures two-way analysis of variance was performed to compare cytokine levels at the three different time points.

Data are presented as mean ± standard deviation or numbers and percentages, as appropriate. Statistical significance was set at *p* < 0.05. All analyses were performed using IBM SPSS Statistics for Windows (ver. 26.0; IBM Corp., Armonk, NY, USA).

## 3. Results

### 3.1. Study Population

A total of 136 patients were assessed for eligibility; 36 patients were excluded and 100 patients underwent randomization. After randomization, four patients (two in group R and two in group D) were excluded because of conversion to laparotomy during surgery. Consequently, 96 patients (48 in each group) were analyzed (Figure 1). The baseline and surgical characteristics of the patients were similar between the two groups (Table 1).

### 3.2. Intraoperative Findings

The intraoperative findings are presented in Table 2. The time to eye opening and that to extubation were comparable between the two groups. Although the incidence of intraoperative bradycardia (HR < 60/min) was comparable between the two groups, that of intraoperative hypotension was higher in group R than in group D (45.8% vs. 22.9%; *p* = 0.018).

### 3.3. Postoperative Pain

Pain scores after 30 min of recovery differed between the groups. Patients in group D reported less pain than those in group R (4.0 ± 1.9 vs. 6.1 ± 2.0, *p* < 0.001, Figure 2). In the recovery room, the proportion of patients who did not require rescue analgesics was higher in group D than in group R (*p* = 0.002, Figure 3). However, there was no significant difference in the postoperative pain at 24 h after surgery (3.2 ± 2.0 in group D vs. 3.3 ± 1.8 in group R, *p* = 0.75).

### 3.4. Other Postoperative Findings

A comparison of the early recovery profiles between the two groups is shown in Table 3. The incidence of analgesic requirement in the recovery room was significantly lower in group D than in group R (25% vs. 66.7%, *p* < 0.001), as well as the incidence of PONV and shivering (2.1% vs. 18.8% and 0% vs. 12.5%, respectively). In addition, the duration of stay in the recovery room was significantly shorter in group D than in group R (33.8 ± 5.0 min vs. 39.2 ± 5.5 min, *p* < 0.001). However, the difference in the early recovery profiles did not last for up to 24 h after surgery; at this time point, the incidence of analgesic requirement, PONV, and shivering was comparable between the groups. Patient-controlled analgesia use and time to flatus were also similar between the two groups (Table 3).

### 3.5. Ancillary Cytokine Study

Table 4 shows a comparison of the MnSOD and MMP 9 levels between the two groups. Both MnSOD and MMP 9 levels showed changes over time. While MnSOD levels showed a gradual reduction over time, MMP-9 levels surged at T1 and decreased at T2 in both groups. However, there were no significant differences between the two groups.

## 4. Discussion

In this study, dexmedetomidine was found to be superior to remifentanil in terms of decreasing pain scores and opioid requirements in the recovery room, as well as in terms of intraoperative hemodynamic stability.

Remifentanil is an ultra-short-acting opioid and a favorable alternative to long-acting opioids during general anesthesia. It is 100–200 times more potent than morphine, with a short elimination half-life and context-sensitive half-life, leading to faster recovery with predictable offset. However, it is also associated with RIH, induced by sensitization of the opioid signaling pathway [13]. RIH increases with increasing remifentanil dose, with higher doses being more susceptible to RIH [14]. Clinically, RIH leads to paradoxical additive requirement of opioids, resulting in opioid-related side effects, such as nausea, vomiting, constipation, respiratory depression, and ileus. Opioid-free anesthesia has been suggested as a method to reduce opioid consumption, and dexmedetomidine has been suggested as an alternative to remifentanil. Dexmedetomidine has been demonstrated to be a reasonable substitute for intraoperative remifentanil in terms of decreasing pain intensity and opioid use [15].

One of the concerns related to dexmedetomidine is hemodynamic instability, such as hypotension and bradycardia. Dexmedetomidine has an interesting biphasic action at different concentrations. High plasma concentrations result in hypertension and bradycardia due to increased systemic vascular resistance, thereby causing baroreceptor stimuli to maintain hemostasis by decreasing the HR. At low plasma concentrations, dexmedetomidine causes hypotension via a vasodilatory effect [16]. In a study comparing the effects of dexmedetomidine and remifentanil for major or intermediate noncardiac surgery, significant bradycardia due to infusion of dexmedetomidine (mean dose 1.2 ± 2 μg/kg/h) resulted in early cessation of the study [17]. Cardiac arrest following high-dose infusion of dexmedetomidine has also been reported [18]. However, our study used a lower dose of dexmedetomidine, with a loading dose of 0.7 ug/kg for 10 min and a maintenance dose within 0.1–1.0 ug/kg/h. The mean dose of dexmedetomidine was 0.6 ± 0.1 μg/kg/h. Additionally, our study population included young, fairly healthy female patients without critical illness, coronary artery disease, and preexisting low blood pressure, all of which are known to be predisposing factors for dexmedetomidine–induced hypotension [19]. This issue about “dose and safety” of dexmedetomidine will be also meaningful in the patients with higher mortality risks in the intensive care unit. Consequently, these imply that an appropriate dose of dexmedetomidine can be safely used in patients undergoing gynecologic laparoscopy.

Dexmedetomidine has demonstrated an opioid-sparing effect in different types of surgical procedures [20,21] by reducing rescue analgesic requirements in the recovery room. In this study, dexmedetomidine resulted in a 35% decrease in postoperative pain scores 30 min after surgery. In addition, the duration of stay in the recovery room was shorter by 5 min in group D. Patients in group R required more opioids as rescue analgesics, which may have led to an increased time of recovery unit stay.

Opioid-induced PONV is an important issue, particularly in gynecological patients. Women are 25 times more likely to experience PONV than men, and gynecological surgery has 6.27 times higher incidences of PONV than other surgical procedures [22]. However, dexmedetomidine has been reported to reduce postoperative nausea (risk ratio (RR) = 0.5–59) and vomiting (RR = 0.48) in various surgical procedures [23,24]. Similar results were obtained in our study. The incidence of PONV at 30 min after surgery was lower in group D.

Dexmedetomidine is known for its anti-shivering effects. Intravenous as well as intrathecal administration of dexmedetomidine decreased postoperative anesthetic shivering in both general and regional anesthesia [9,17,25]. In our study, dexmedetomidine had a significantly better postoperative anti-shivering effect than remifentanil; in fact, none of the patients in group D experienced post-anesthetic shivering compared to those in group R.

The oxidative stress theory has been suggested as the underlying mechanism of RIH. In some experimental and clinical studies, high doses of remifentanil have been reported to cause significant increases in oxidative stress markers, such as malondialdehyde, 3-nitrotyrosine, or MMP-9, and decrease in oxygen radical scavengers, such as MnSOD [4]. Recently, dexmedetomidine has been reported to decrease the level of reactive oxygen species in animal studies [26,27,28,29,30]. To determine the effect of dexmedetomidine on oxidative stress compared to that of remifentanil, we measured MMP-9 and MnSOD levels. However, our study failed to prove the effects of dexmedetomidine on changes in MnSOD and MMP-9 levels. The most likely reason may be the difference in the methods used to collect and measure the cytokines. Although a previous animal study measured cytokines collected directly from the spinal cord [4], our study used blood samples because of ethical concerns. This difference in sampling methods may have led to the dilution of cytokines in the blood.

Our study had some limitations. First, we only included patients undergoing gynecological laparoscopy, which may interfere with the generalizability of our results to other populations. However, because we chose a population at risk of PONV, our findings suggest a beneficial effect of dexmedetomidine in terms of antiemetic effects in other surgical procedures or populations with a high risk of PONV. Second, because of the different infusion protocols, anesthesiologists participating in the intraoperative management were aware of the group allocation. However, caregivers and assessors were blinded to the group allocation.

## 5. Conclusions

In the present study, low-dose dexmedetomidine decreased pain severity after gynecologic laparoscopy and opioid-induced side effects in the immediate postoperative period without accompanying intraoperative hemodynamic instability. Further studies with more elaborate methods are required to demonstrate the antioxidative effects of dexmedetomidine in clinical settings.

## Figures and Tables

**Figure 1 jcm-12-00350-f001:**
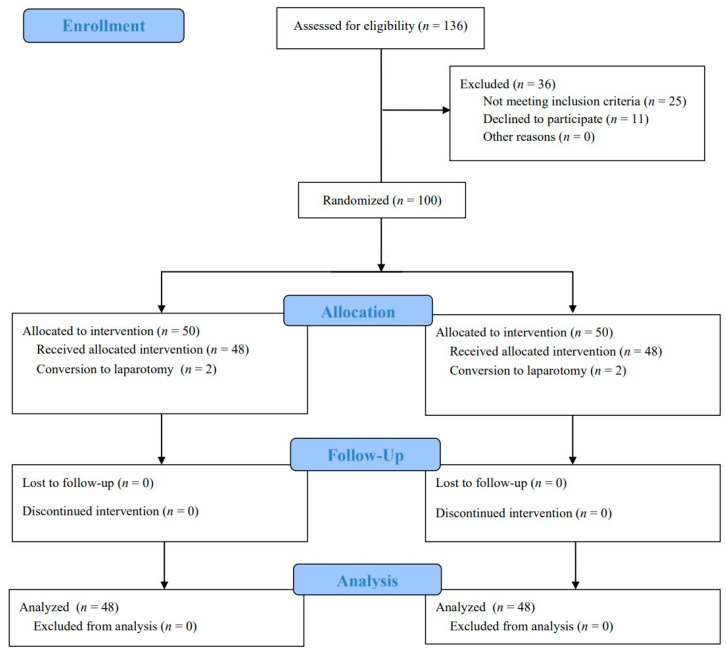
CONSORT 2010 flow diagram.

**Figure 2 jcm-12-00350-f002:**
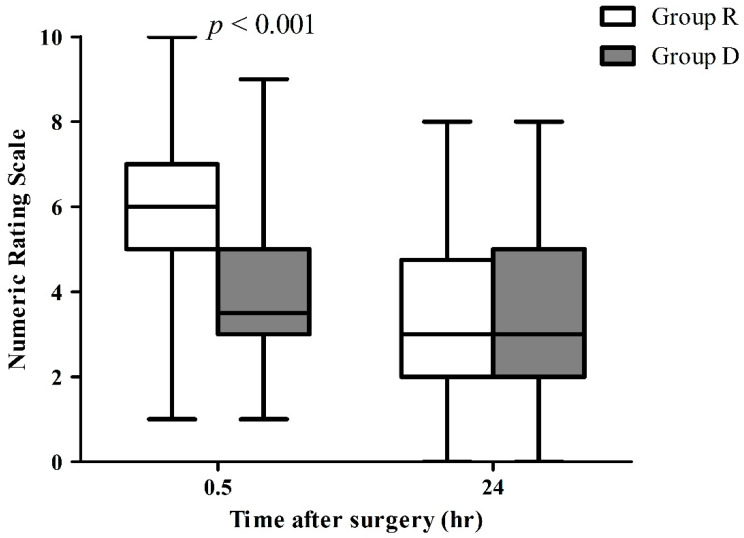
Postoperative pain difference between the two groups. Box plot with median (solid line), 25–75th percentiles (box), and min-max (whiskers). Mann–Whitney’s *U* test was used.

**Figure 3 jcm-12-00350-f003:**
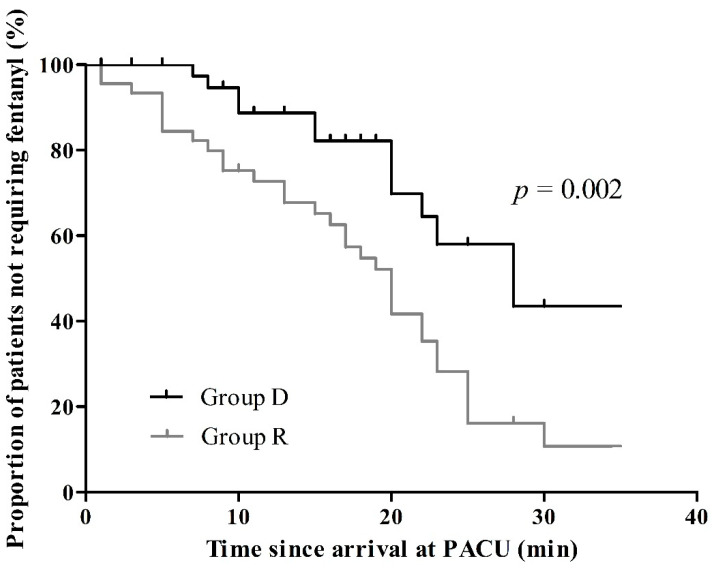
Opioid requirement difference between the two groups. The vertical upticks indicate the occurrence of events (opioid requirement). Kaplan–Meier analysis was performed. PACU, Post-anesthetic care unit.

**Table 1 jcm-12-00350-t001:** Characteristics of the patients at baseline.

	Group R	Group D	*p* Value
	(*n* = 48)	(*n* = 48)	
Age	43.9 ± 10.7	41.7 ± 9.5	0.28
Height (cm)	159.0 ± 5.7	160.5 ± 4.9	0.16
Weight (kg)	58.5 ± 10.1	58.3 ± 7.2	0.90
ASA classification (I/II)	34/14	39/9	0.23
Type of surgery			0.15
Hysterectomy ^a^	29 (60.5%)	25 (52.1%)	
Myomectomy ^b^	4 (8.3%)	6 (12.5%)	
Cystectomy/Cyst enucleation only	10 (20.8%)	15 (31.2%)	
Adnexectomy only	5 (10.4%)	2 (4.2%)	

Values are expressed as mean ± SD and numbers (proportion). ASA; American Society of Anesthesiologists. ^a^ with or without adnexectomy. ^b^ with or without cystectomy/cyst enucleation.

**Table 2 jcm-12-00350-t002:** Intraoperative findings.

	Group R (*n* = 48)	Group D (*n* = 48)	*p* Value
Anesthetic duration (min)	148.0 ± 51.2	141.4 ± 5.4	0.47
Time to eye opening (min)	5.1 ± 1.7	5.1 ± 1.8	0.93
Time to extubation (min)	5.7 ± 1.8	5.7 ± 2.0	0.95
Intraoperative hypotension	22 (45.8%)	11 (22.9%)	0.018
Intraoperative bradycardia	7 (14.6%)	3 (6.3%)	0.18
Remifentanil dose (μg/kg/min)	0.1 ± 0.04		
Dexmedetomidine dose (μg/kg/h)		0.6 ± 0.1	

Values are expressed as mean ± SD or numbers (proportion).

**Table 3 jcm-12-00350-t003:** Postoperative findings.

	Group R (*n* = 48)	Group D (*n* = 48)	*p* Value
0.5 h after surgery			
Rescue analgesic use	32 (66.7%)	12 (25%)	<0.001
PONV	9 (18.8%)	1 (2.1%)	0.008
Rescue antiemetic use	32 (66.7%)	12 (25%)	<0.001
Sedation level (level 1/2/3)	44/3/1	39/9/0	0.1
Shivering	6 (12.5%)	0 (0%)	0.01
Time of PACU stay (min)	39.2 ± 5.5	33.8 ± 5.0	<0.001
24 h after surgery			
PCA use (ml)	17.7 ± 14.1	17.2 ± 11.1	0.9
Rescue analgesic use	15 (31.3%)	13 (27.1%)	0.8
PONV	13 (27.1%)	17 (36.4%)	0.5
Rescue antiemetic use	3 (6.3%)	2 (4.2%)	>0.999
Shivering	7 (14.6%)	6 (12.5%)	>0.999
Pruritus	1 (2.1%)	1 (2.1%)	>0.999
Time to first flatus (h)	20.1 ± 6.2	22.5 ± 5.8	0.06

Values are expressed as mean ± SD and numbers (proportion). Abbreviations: NRS; Numeric Rating Scale, PONV; Postoperative nausea and vomiting, PACU; Post-anesthetic care unit, PCA; Patient controlled analgesics. Sedation level: 1 = awake, 2 = sedated and responsive to verbal stimuli, 3 = sedated and unresponsive to verbal stimuli.

**Table 4 jcm-12-00350-t004:** Ancillary cytokine levels.

	Group R	Group D	*p* Value
	(*n* = 20)	(*n* = 20)	
MnSOD (ng/mL)			0.5
T0	0.6 ± 0.4	0.6 ± 0.2	
T1	0.6 ± 0.2	0.6 ± 0.2	
T2	0.4 ± 0.2	0.5 ± 0.2	
MMP-9 (pg/mL)			0.6
T0	187.3 ± 85.7	191.5 ± 100.9	
T1	385.1 ± 211.3	352.4 ± 230.2	
T2	331.9 ± 139.6	351.6 ± 209.4	

Values are expressed as mean ± SD. Abbreviations: MnSOD; Manganese superoxide dismutase, MMP-9; Matrix metalloproteinase. T0; before surgery, T1; 30 min after surgery, T2; 24 h after surgery.

## Data Availability

The data generated in this study can be shared after a reasonable request to the corresponding author.

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
