# Peer review of "Efficacy of Dexmedetomidine vs. Remifentanil for Postoperative Analgesia and Opioid-Related Side Effects after Gynecological Laparoscopy: A Prospective Randomized Controlled Trial"

_jcm, 2023, doi:10.3390/jcm12010350_

Round 1

Reviewer 1 Report

The study is fair with a good protocol.

Regarding the mentioned article here are my arguments;

- the study from the scientific point of view is not a novelty, opioid free anesthesia being very much discussed   

BUT

- it has a correct design, very explicit from the point of view of the methodology

- the discussion as well as the conclusions answer the aim of the study

- it expresses a clinical experience   

Comments:

- exclusion criteria: include ASA

- line 101 - clarify: "with SBT maintained within 20% of baseline...

- line 116 - clarify: basal rate.....

Reviewer 2 Report

This is a randomized controlled trial on the difference between remifentanil and dexmedetomidine on opioid related side effects. they randomized 50 women to receive remifentanil and 50 to dexmedetomidine. They found lower pain scores and fewer opioid related side effects in the dexmedetomidine group. 

The results are as expected, because the effects of remifentanil is gone within 5-10 minutes after stopping the infusion and dexmedetomidine has a longer context sensitive half life. As a consequence, pain scores were higher, more analgesics were requrested and more PONV was reported (due to rescue analgesics) in the remifentanil group.

The subject that dexmedetomine can have serious hemodynamic effects was touched in the discussion, though could be elaborated since a statement was issued with a higher mortality risk in patients < 65 years of age admitted to the ICU (non-surgical) 

Furthermore, I am interested in the hemodymic intraoperative data. They used a 'low' dose infusion of dexmedetomidine. Was this enough for blocking the surgical stimulus. especially the pneumoperitoneum? Were there any changes in blood pressure between the groups? I am curious about the study algorithm if blood pressure rose above 20% of baseline.
